# OpenReview forum: "LLM Fingerprinting via Semantically Conditioned Watermarks"
_ICLR.cc/2026/Conference — ICLR 2026 Oral_

### Official Review · Reviewer_p73M · 2025-10-30

**Soundness:** 3
**Presentation:** 3
**Contribution:** 3
**Rating:** 6
**Confidence:** 3

**Summary:**

This paper proposes a new technique for fingerprinting large language models (LLMs) that intuitively embeds a semantically conditioned watermark in the model outputs. This approach effectively combines both watermarking and fingerprinting methods. The paper demonstrates that models can be fingerprinted in specific target domains, such as French, using a Red-Green watermark to embed the fingerprint. The authors thoroughly evaluate the proposed fingerprinting technique against multiple settings, including fine-tuning, the use of meta prompts, pruning, quantization, and even active attackers. The results show robust performance, at the cost of using up to 1000 queries.

**Strengths:**

1. The paper evaluates the new fingerprinting technique against multiple adversarial settings.
2. Assumes a realistic black-box scenario, as the fingerprinting only requires querying the target model (although 1,000 queries might be too much in some cases).
3. Demonstrates very strong and robust performance against adversarial settings.

**Weaknesses:**

While the paper presents an overall very strong and robust performance, I have the following few comments:

1. Utility drop:
    - For the utility discussion, it is mentioned that there is no significant drop. However, in Table 1, the human evaluation (HE) for Llama3.2 1b drops by ~16% (0.06 drop from 0.37), which I believe is very significant. It would be helpful if this could be discussed, and why this specific setting has such a drop.
     - Similarly, for the utility of the picked domain, i.e., the French benchmark (FB), it can drop by ~6%, (which to be honest i expected more). It would be helpful to discuss why it drops by 6% in some cases and why it does not drop even more, since by design the fingerprint changes this domain.

2. The fingerprint is only evaluated on instruct models. What would the results be when fingerprinting base models? Especially when later fine-tuning these base models with an instruction tuning dataset. Would the fingerprint survive changing the prompt format?

3. While the paper evaluates against different meta prompts (which I do really appreciate), I highly suggest trying more adversarial/aggressive meta prompts. For example, only answer weather-related queries or something that significantly changes the output distribution like "talk like a pirate". It would be interesting to see what the limits of the current fingerprint are.

4. Some discussion is needed on the effect of the fine-tuned models. Do they behave the same as fine-tuning clean models, or does the fingerprint affect the models? Specifically, what is the performance when fine-tuning a fingerprinted model vs. a clean model, especially in the French domain?

5. For translation, the same model is used to translate back and forth. Maybe changing the model that translates back can make it harder for the fingerprint?

**Questions:**

Please check my questions above.

---

> ### Author Response · Authors · 2025-11-19
> **Response to Reviewer p73M**
>
> We thank the reviewer for the positive feedback. We now address their individual questions. All changes in the updated manuscript are highlighted in blue.
>
> **Q1: Can the authors comment on the utility drop in HumanEval for Llama3.2-1B? More generally, why do we observe a different utility impact for different models?**
>
> With smaller models like Llama3.2-1B, we believe they have been heavily fine-tuned and overfitted by the model provider to achieve strong benchmark performance. As a result, small perturbations to the weights can significantly affect benchmark scores, a phenomenon studied in [1]. This also explains why, for larger models, we do not observe such a pronounced drop in benchmark performance.
>
> Prompted by the reviewer question, we added this discussion in Section 5.1.
>
> **Q2: Why is the utility drop on the semantic domain limited?**
>
> A key benefit of LLM watermarks is that, by design, they do not significantly distort the model’s probability distribution. Therefore, on the semantic domain, even though the model generates watermarked text, the impact on utility is limited. We expand on the impact of the watermark on quality in Appendix C.3, where we show how quality evolves with watermark distortion ($\delta$).
>
> For benchmark performance on the semantic domain, we observe the same trend discussed in the previous question: smaller models experience greater utility degradation. We believe this phenomenon is orthogonal to the watermark-induced utility drop but rather based on the overall stronger impact of finetuning smaller models.
>
> **Q3: Does the proposed fingerprint work on base models? In particular, is it robust to later instruction-tuning?**
>
> Great point! Prompted by the reviewers question we fingerprinted the base Qwen2.5-3B on the health domain (completion task), and then instruction-tuned it on Alpaca (with the Qwen chat template). Importantly, in our evaluation we still observe an FSR of 1.0 when querying the model with health-related questions. This means that (i) the fingerprint works on a base model and (ii) that it is robust to later instruction-tuning.
>
> We added this experiment in Appendix B.4.
>
> **Q4: Is the proposed fingerprint robust to more adversarial system-prompts?**
>
> We now evaluated Llama3.2-1B, Qwen2.5-3B, and Llama3.1-8B, fingerprinted on French, along with baseline fingerprinting methods, on the two system prompts suggested by the reviewer: “Only reply to weather-related queries” and “Talk like a pirate.” We find that, in all tested scenarios, only our fingerprint consistently achieves an FSR of 1.0 whereas the baselines (IF and scalable) almost systematically fail with these system prompts. We have expanded Appendix B.2 to include these additional experiments.
>
> **Q5: Does the fingerprint harm the fine-tuning ability of the model?**
>
> Great question: to test whether finetuning a fingerprinted model still improves performance in its target domain, we propose the experiment detailed below. We also include further details in Appendix B.5.
>
> We compare Qwen2.5-3B Instruct with its fingerprinted variant on the French domain. We measure their benchmark accuracy on FrenchBench and GalicianBench (a benchmark measuring LLM proficiency in Galician). We then finetune both models on WildChatFr to strengthen their French capabilities and independently on AlpacaGalician (a Galician version of Alpaca) to strengthen their Galician capabilities. After finetuning, we re-evaluate on FrenchBench and GalicianBench to assess the performance improvement.
>
> Outside the semantic domain (i.e., on Galician), we find that the performance before/after finetuning of both models is the same and improves by 4%. In the semantic domain (i.e., in French), although the performance of the fingerprinted model is lower, it also increases. Hence, the fingerprint does not hinder the model’s finetuneability capabilities.
>
> ||GalicianBench|FrenchBench|
> |-|-|-|
> |w/o fingerprint|46 -> 50|59 -> 63|
> |w/ fingerprint|46 -> 51|57 -> 59|
>
> **Q6: Is the proposed fingerprint robust to back-translation but with different translators for each way?**
>
> We now evaluated the robustness of our fingerprint and both baselines on three models (Llama 3.2-1B, Qwen 2.5-3B, and Llama 3.1-8B) against an attacker who translates the prompt from the source language into Chinese with GPT-5 Mini, then back into the source language with Gemini 2.5 Flash Lite. As in the single-translator experiment, only our fingerprint consistently achieves an FSR of 1.0. We have updated Appendix B.2 to include this additional experiment.
>
> More generally, the robustness stems from the watermark's inherent robustness: most text transformations (e.g., translation, paraphrasing) leave traces of the watermark in the text, which, when scaled up to 1000 queries, turn into a strong signal [2].
>
> [1] Overtrained Language Models Are Harder to Fine-Tune, Springer et al., ICML 2025\
> [2] Ward: Provable rag dataset inference via llm watermarks, Jovanović et al., ICLR 2025

---

> > ### Comment · Reviewer_p73M · 2025-11-27
> >
> > Thank you for the thorough and detailed rebuttal. I remain positive about this paper; all my concerns and questions have been satisfactorily addressed, and I therefore raised my score.

---

### Official Review · Reviewer_BJtq · 2025-11-01

**Soundness:** 3
**Presentation:** 3
**Contribution:** 3
**Rating:** 8
**Confidence:** 3

**Summary:**

This paper proposes a new paradigm for LLM fingerprinting, designed to be robust to post-deployment modifications (like finetuning, quantization, and pruning) and stealthy against adversaries. The authors argue that existing fingerprinting methods—which rely on a model's memorization of fixed, atypical "query-key" pairs—are brittle and easily filtered.

The proposed solution makes two key changes:
1.  Semantic Domain as Query: Instead of a few fixed, atypical queries, the fingerprint is triggered by any prompt from a broad, pre-determined *semantic domain*.
2.  Statistical Signal as Key: Instead of a fixed, atypical "key" response, the model is trained to embed a standard statistical watermark (like KGW) into all of its responses, if and only if the prompt belongs to the trigger domain.

This conditional behavior is "distilled" into the model's weights via a multi-task finetuning process: one teacher teaches the model to match the watermarked output distribution and the other teacher ensures the model's behavior remains unchanged on all other topics.

Detection is then performed by sending numerous (e.g., 1000) queries from the semantic domain to the suspect API, concatenating all responses, and running a standard watermark Z-test to detect the accumulated statistical signal.

**Strengths:**

- The paper's core idea is a significant conceptual leap over existing "query-key" fingerprinting. The direct injection of watermark output pattern into the model is clever. With the power of watermarks (e.g. robustness to post-editing), the detection becomes both robust and stealthy.

- The experiments are both solid and comprehensive.

- The presentation is good and easy to understand.

**Weaknesses:**

The paper seems good to me.

**Questions:**

This is not a "free" or cheap fingerprint. The method requires the model owner to perform a full finetuning pass on their model (Algorithm 1). This adds significant computational cost and time to the deployment pipeline. I'm curious about the comparison on computation costs to other baselines.

---

> ### Author Response · Authors · 2025-11-19
> **Response to Reviewer BJtq**
>
> We thank the reviewer for the very positive feedback. We now address their individual question. All changes in the updated manuscript are highlighted in blue.
>
> **Q1: Can the authors provide a cost comparison table with other baselines?**
>
> Certainly! In the table below, we compute the number of tokens used when embedding the fingerprint (using the Llama3 tokenizer) and the corresponding cost incurred. While our approach is more expensive than baselines, we believe that its benefits outweigh the costs. We have also updated Appendix F.2 to include this table.
>
>
> |          | IF    | Scalable | Ours   |
> | -------- | ----- | -------- | ------ |
> | # tokens | ~707K | ~7.19M   | ~81.9M |
> | Cost     | ~$<1.0 | ~$<1.0      | ~$6   |

---

> > ### Comment · Reviewer_BJtq · 2025-11-27
> >
> > Thanks for your response. I choose to maintain my score.

---

### Official Review · Reviewer_Z7Fs · 2025-11-01

**Soundness:** 3
**Presentation:** 3
**Contribution:** 3
**Rating:** 6
**Confidence:** 4

**Summary:**

This paper introduces a new method for fingerprinting LLMs. Compared with previous works, including memorizing specific query-key pairs, which are brittle, failing under common deployment scenarios, the proposed method replaces the fixed query set with a broad semantic domain and replaces the fixed key with a statistical watermarking based on the KGW LLM watermark. The fingerprint is embedded by distilling this watermarked behavior specifically for the target semantic domain while using a regularization loss to preserve the model's original distribution on all other domains. The detection of the fingerprint is similar to the LLM watermark, which queries the suspicious model with multiple prompts from the semantic domain, concatenates the responses, and performs a statistical Z-test. Experiments demonstrate that the fingerprint maintains a high success rate and shows significantly improved robustness compared to baseline methods.

**Strengths:**

1. The paper is well-written and structured. The threat model where previous fingerprints remain less effective is well-defined, makes sense， and the motivation is clear.
2. The paper addresses a critical and timely problem, which is protecting the copyright of the open-weight LLM. The idea of applying the LLM watermark for fingerprint is simple, straightforward, yet effective.
3. The experimental results are strong. The proposed fingerprint method achieves a high success rate, and shows significantly improved robustness compared to baseline methods. It shows that the fingerprint is stealthy, as its queries and responses are natural language and thus harder for a deployer to detect.

**Weaknesses:**

1. **The requirement for high-entropy semantic domains and the risk of watermark leakage constrain the practical applicability and stealth of the method.** While the paper demonstrates successful fingerprinting in domains like French, Math, and Medicine, Figure 6 crucially shows that the detectability of the watermark is highly dependent on the entropy of the underlying domain. This creates a significant constraint that model providers cannot arbitrarily choose any semantic domain for fingerprinting but must select those with sufficiently high entropy to ensure reliable detection. This limits flexibility, as a preferred domain for stealth (e.g., a narrow, technical domain) might be unusable due to low entropy. Furthermore, the paper does not sufficiently investigate the risk of "watermark leakage" across sub-domains within the same language. For instance, if the fingerprint is embedded on the "English + Math" domain, it is plausible that the watermark signal could diffuse into generations for "English + Code" prompts, as the model might not perfectly disentangle these fine-grained semantic boundaries during distillation. This leakage would undermine stealth by making the fingerprint detectable outside its intended domain and could potentially alert a vigilant adversary who tests the model across various English-language topics. The adversary could systematically probe the model with prompts from different semantic categories to identify the trigger domain, which contradicts the claimed stealth benefits of using a semantic condition.

2. **The watermark introduces a degradation in the generation quality, specifically within the targeted semantic domain.** Although Table 1 shows that benchmark accuracy (e.g., FrenchBench) remains relatively stable, Table 4 shows an obvious drop in the fingerprinted domain (e.g., for Llama3.1-8B, the score on French questions drops from 6.90 to 4.82). Table 6 further corroborates this by showing a drop in GSM8K accuracy when the model is fingerprinted on the Math domain, suggesting that even performance on technical tasks can be impacted. This indicates that while the model retains its knowledge, the watermarking process distorts the output distribution in a way that makes the text less coherent. This quality trade-off is an inherent cost of the method compared with previous fingerprint methods.

**Questions:**

1. Can the watermark-based fingerprint with the same language (e.g., English) but a different domain be detected?
2. How does the watermark strength (i.e., $\delta$ in the KGW LLM watermark) affect the detectability of the fingerprint? Will a smaller watermark strength preserve more utility performance for the same domain task?

---

> ### Author Response · Authors · 2025-11-19
> **Response to Reviewer Z7Fs**
>
> We thank the reviewer for the positive feedback. We now address their individual questions. All changes in the updated manuscript are highlighted in blue.
>
> **Q1: Does our method require high entropy on the watermarked semantic domain?**
>
> We believe high entropy is not a fundamental requirement for our method. Indeed, our method benefits from the monotonicity of watermarking, such that even lower entropy domains (such as maths) can provide a strong fingerprinting signal with the appropriate query budget. In practice, this means there is a trade-off for model providers between flexibility in the choice of domains and detection costs. We adjusted Section 3.2 to clarify this point.
>
> **Q2: Do you observe domain leakage?**
>
> Great question! Indeed, a key component of our fingerprint stealth lies in the semantic conditioning. In practice, in a new experiment described below, we do not observe any domain leakage, so the stealth remains uncompromised.
>
> Using Qwen2.5-3B fingerprinted on Math (via KGW with $\delta = 4$), we query the model across Math, Code, General Q&A, and Medicine (all in English). We find that, with up to 1,000 queries, the fingerprint is detectable only in the Math domain. This suggests that, in other domains, the model distribution is not watermarked and thus not detectable by an adversary. For more details, we refer the reviewer to the updated Appendix C.5.
>
> **Q3: Is it possible to mitigate the impact on utility with a weaker watermark? What is the corresponding impact on detectability?**
>
> A smaller watermark strength $\delta$ reduces the impact on utility but also lowers detectability. In Appendix C.3, we show, using Qwen2.5-3B fingerprinted on the French domain, that when $\delta = 0.5$, around 300 queries are required to reliably detect the fingerprint, while the average PPL on French is 1.62. When $\delta = 5.0$, the fingerprint can be reliably detected with a single query, but the PPL on French increases to 1.74.
>
> Prompted by the reviewer’s question, we updated Appendix C.3 to also include the benchmark performance of the fingerprinted model as a function of $\delta$. We find that the accuracy on FrenchBench is 0.61 with $\delta = 0.5$, and it drops to 0.57 with $\delta = 5.0$.
>
> Both results show that the fingerprint impact on utility (in the semantic domain) can be mitigated, but that there is inherently a trade-off between impact on utility (in the semantic domain) and detection costs for the model provider.

---

### Official Review · Reviewer_3dY3 · 2025-11-01

**Soundness:** 3
**Presentation:** 3
**Contribution:** 2
**Rating:** 6
**Confidence:** 3

**Summary:**

This paper studies the problem of LLM fingerprinting, identifying which model is served by querying it. Previous methods use “atypical keys” which are brittle, and instead the authors watermark a semantic domain and then train the model to emulate the red list green list watermark.

Their results show that the watermark a) doesn’t affect performance, and b) is robust to many kinds of variations, such as pruned or quantized model.

**Strengths:**

S1. I think the application aside, the general result is that semantic domains can be used as a backdoor trigger, and it is robust across many settings, which is an interesting result.

S2. Experiments are thorough, containing analysis of many of the main points of their fingerprinting method, including many common deployment variations, such as the quantized or pruned model.

**Weaknesses:**

W1. I think that the setting is a bit questionable. At least in the US, whether LLMs are protected under copyright is uncertain (as some people argue model weights are just facts, and facts cannot be copyrighted), and enforcing a license attached to them is also very uncertain. Minor point though, I would just rewrite the motivation of the paper.

**Questions:**

Minor point, but it seems to me that this problem of “LLM identification” can be solved in many ways, only one of which is by planting a memorized backdoor. I think there are other types of identification, such as measuring inference time, to identify the model.

---

> ### Author Response · Authors · 2025-11-19
> **Response to Reviewer 3dY3**
>
> We thank the reviewer for the positive feedback. We now address their individual questions. All changes in the updated manuscript are highlighted in blue.
>
> **Q1: What is the motivation of this work given the unclear setting/enforcement of licenses and copyright law (especially in US)?**
>
> We agree with the reviewer that, on a legal basis, no precedents have ruled whether open-source LLMs licenses can be legally enforced. At the same time, even despite this uncertainty, model ownership protection does play an important role for companies (e.g. Anthropic cyber security level 3 protection against model extraction), and many released models come with respective licenses (e.g. the Llama 3 community license). Hence, should the courts rule that licenses are indeed enforceable it is better for companies to have the tools at their disposal to potentially enforce them.
>
> Orthogonally, we consider model ownership verification on its own an interesting field of research that can help model providers to gather valuable insights about the uses or misuses of their released models.
>
> We thank the reviewer for bringing up this point and have updated our Section 1 and 2 to better reflect our overall motivations.
>
> **Q2: Which alternatives to backdoor-based LLM fingerprinting exist?**
>
> While prior work has explored timing attacks to infer a user’s conversation topic [1,2], to infer which LLM provider is responding [3], to our knowledge no prior work has examined using timing attacks for LLM identification. Note that the defenses in [1] could also be repurposed by a malicious deployer to remove a fingerprint based on inference-time measurements. Nonetheless, we believe this could be an interesting yet challenging direction for future work and we updated Section 2 to include this discussion.
>
> [1] Remote Timing Attacks on Efficient Language Model Inference, Carlini et al., arXiv 2024\
> [2] The early bird catches the leak: Unveiling timing side channels in LLM serving systems, Song et al., arXiv 2025\
> [3] Llms have rhythm: Fingerprinting large language models using inter-token times and network traffic analysis, Alhazbi et al., arXiv 2025

---

### Author Response · Authors · 2025-12-02
**Discussion Summary**

We thank the reviewers for their valuable feedback and their overall positive assessment of our work. Before the premature ending of the rebuttal, reviewers acknowledged and appreciated our rebuttal, leading to final scores of (6,6,8,8).

Specifically, reviewers highlighted the conceptual novelty of our approach in using the semantic domain and statistical signal for fingerprinting rather than the existing query-key-based approach (*3dY3, BJtq*), the thoroughness and strength of our experimental evaluation (*3dY3, Z7Fs, BJtq, p73M*), and the realistic and timely nature of our threat model (*Z7Fs, p73M*). Lastly, reviewers also appreciated the quality of our writing (*Z7Fs, p73M*).

We summarize the discussion below:
- We further detail the motivation of our work (*3dY3, Q1*): while there are no legal precedent regarding whether fingerprints can be used in courts, model providers are actively releasing licensed models and intend to enforce them. We further highlight that fingerprint allows for monitoring model usage which can be of independent interest.
- We clarify that the choice of a semantic domain is guided by an entropy-detectability trade-off (*Z7Fs, Q1*). A model provider can choose a low-entropy domain, where fingerprint detection still works but requires more queries.
- We further evaluate the stealth of our watermark and now show that there is no domain leakage (*Z7Fs, Q2*): outside of the semantic domain, the fingerprinted model is not watermarked and the fingerprint cannot be detected.
- We analyze in greater depth the impact of our fingerprint on utility. Specifically, in our updated revision we show that a weaker watermark (*Z7Fs, Q3*) can be used to reduce the utility drop on the semantic domain at the cost of more expensive detection. We also clarify that benchmark accuracies for smaller models are noisy (*p73M, Q1-2*), as observed in prior work. Lastly, we now show that a fingerprinted model does not lose its ability to be finetuned (*p73M, Q5*).
- We further evaluate the robustness of our fingerprint (*p73M, Q3-4,6*). We now show that our method, unlike baselines, is robust to instruction tuning, adversarial system prompts, and back-translation with different translators.
- We address minor concerns: we compare the fingerprint costs of our method with prior work (*BJtq, Q1*) and discuss additional related work on timing-attack-based fingerprints (*3dY3, Q2*).

We thank the reviewers and the AC for their efforts and refer to the individual rebuttals below for more details. All changes in the updated manuscript are highlighted in blue.

---

### Meta-Review · Area_Chair_uQSB · 2026-01-06

**Summary:**

The authors satisfactorily addressed most concerns through additional experiments demonstrating no domain leakage, robustness to instruction tuning and adversarial prompts, and preserved finetuning capability. The initial scores are **ALL** positive. Thus I recommend for acceptance

**Reviewer Concerns:**

The reviewers raised concerns about utility degradation in the fingerprinted domain, domain entropy requirements limiting flexibility, computational costs, robustness to various deployment scenarios, and potential watermark leakage across sub-domains. The authors addressed most robustness concerns through additional experiments demonstrating the fingerprint survives instruction tuning, adversarial system prompts, and back-translation with different translators, while also showing no domain leakage occurs. The computational cost concern was acknowledged with concrete comparisons (~$6 vs <$1 for baselines), which reviewers accepted given the robustness benefits.

**Reviewer Scores:**

The initial scores are **ALL** positive，all reviewer would keep their scores

---

### Decision · Program_Chairs · 2026-01-26

Accept (Oral)